# Binary Classification from Positive-Confidence Data

**Takashi Ishida**[1,2]    **Gang Niu**[2]    **Masashi Sugiyama**[2,1]
[1] The University of Tokyo, Tokyo, Japan
[2] RIKEN, Tokyo, Japan
{ishida@ms., sugi@}k.u-tokyo.ac.jp, gang.niu@riken.jp

## Abstract

Can we learn a binary classifier from only positive data, without any negative data or unlabeled data? We show that if one can equip positive data with confidence (*positive-confidence*), one can successfully learn a binary classifier, which we name *positive-confidence (Pconf) classification*. Our work is related to *one-class classification* which is aimed at "describing" the positive class by clustering-related methods, but one-class classification does not have the ability to tune hyper-parameters and their aim is not on "discriminating" positive and negative classes. For the Pconf classification problem, we provide a simple empirical risk minimization framework that is model-independent and optimization-independent. We theoretically establish the consistency and an estimation error bound, and demonstrate the usefulness of the proposed method for training deep neural networks through experiments.

## 1 Introduction

Machine learning with big labeled data has been highly successful in applications such as image recognition, speech recognition, recommendation, and machine translation [14]. However, in many other real-world problems including robotics, disaster resilience, medical diagnosis, and bioinformatics, massive labeled data cannot be easily collected typically. For this reason, machine learning from *weak supervision* has been actively explored recently, including *semi-supervised classification* [6, 30, 40, 53, 23, 36], *one-class classification* [5, 21, 42, 51, 16, 46], *positive-unlabeled (PU) classification* [12, 33, 8, 9, 34, 24, 41], *label-proportion classification* [39, 54], *unlabeled-unlabeled classification* [7, 29, 26], *complementary-label classification* [18, 55, 19], and *similar-unlabeled classification* [1].

In this paper, we consider a novel setting of classification from weak supervision called *positive-confidence (Pconf) classification*, which is aimed at training a binary classifier only from positive data equipped with *confidence*, without negative data. Such a Pconf classification scenario is conceivable in various real-world problems. For example, in purchase prediction, we can easily collect customer data from our own company (positive data), but not from rival companies (negative data). Often times, our customers are asked to answer questionnaires/surveys on how strong their buying intention was over rival products. This may be transformed into a probability between 0 and 1 by pre-processing, and then it can be used as positive-confidence, which is all we need for Pconf classification.

Another example is a common task for app developers, where they need to predict whether app users will continue using the app or unsubscribe in the future. The critical issue is that depending on the privacy/opt-out policy or data regulation, they need to fully discard the unsubscribed user's data. Hence, developers will not have access to users who quit using their services, but they can associate a positive-confidence score with each remaining user by, e.g., how actively they use the app.

In these applications, as long as positive-confidence data can be collected, Pconf classification allows us to obtain a classifier that discriminates between positive and negative data.

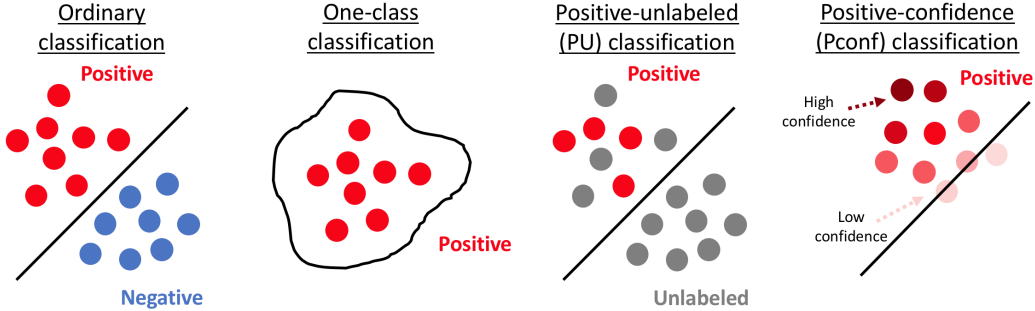

**Figure 1:** Illustrations of the *Pconf classification* and other related classification settings. Best viewed in color. Red points are positive data, blue points are negative data, and gray points are unlabeled data. The dark/light red colors on the rightmost figure show high/low confidence values for positive data.

**Related works**  Pconf classification is related to one-class classification, which is aimed at "describing" the positive class typically from hard-labeled positive data without confidence. To the best of our knowledge, previous one-class methods are motivated geometrically [50, 42], by information theory [49], or by density estimation [4]. However, due to the descriptive nature of all previous methods, there is no systematic way to tune hyper-parameters to "classify" positive and negative data. In the conceptual example in Figure 1, one-class methods (see the second-left illustration) do not have any knowledge of the negative distribution, such that the negative distribution is in the lower right of the positive distribution (see the left-most illustration). Therefore, even if we have an infinite number of training data, one-class methods will still require regularization to have a tight boundary in all directions, wherever the positive posterior becomes low. Note that even if we knew that the negative distribution lies in the lower right of the positive distribution, it is still impossible to find the decision boundary, because we still need to know the degree of overlap between the two distributions and the class prior. One-class methods are designed for and work well for anomaly detection, but have critical limitations if the problem of interest is "classification".

On the other hand, Pconf classification is aimed at constructing a discriminative classifier and thus hyper-parameters can be objectively chosen to discriminate between positive and negative data. We want to emphasize that the key contribution of our paper is to propose a method that is purely based on empirical risk minimization (ERM) [52], which makes it suitable for binary classification.

Pconf classification is also related to positive-unlabeled (PU) classification, which uses hard-labeled positive data and additional unlabeled data for constructing a binary classifier. A practical advantage of our Pconf classification method over typical PU classification methods is that our method does not involve estimation of the *class-prior probability*, which is required in standard PU classification methods [8, 9, 24], but is known to be highly challenging in practice [44, 2, 11, 29, 10]. This is enabled by the additional confidence information which indirectly includes the information of the class prior probability, bridging class conditionals and class posteriors.

**Organization**  In this paper, we propose a simple ERM framework for Pconf classification and theoretically establish the consistency and an estimation error bound. We then provide an example of implementation to Pconf classification by using linear-in-parameter models (such as Gaussian kernel models), which can be implemented easily and can be computationally efficient. Finally, we experimentally demonstrate the practical usefulness of the proposed method for training linear-in-parameter models and deep neural networks.

## 2 Problem formulation

In this section, we formulate our Pconf classification problem. Suppose that a pair of $d$-dimensional pattern $\boldsymbol{x} \in \mathbb{R}^d$ and its class label $y \in \{+1, -1\}$ follow an unknown probability distribution with density $p(\boldsymbol{x}, y)$. Our goal is to train a binary classifier $g(\boldsymbol{x}) : \mathbb{R}^d \to \mathbb{R}$ so that the classification risk $R(g)$ is minimized:

$$R(g) = \mathbb{E}_{p(\boldsymbol{x},y)}[\ell(yg(\boldsymbol{x}))], \tag{1}$$

where $\mathbb{E}_{p(\boldsymbol{x},y)}$ denotes the expectation over $p(\boldsymbol{x}, y)$, and $\ell(z)$ is a loss function. When margin $z$ is small, $\ell(z)$ typically takes a large value. Since $p(\boldsymbol{x}, y)$ is unknown, the ordinary ERM approach [52] replaces the expectation with the average over training data drawn independently from $p(\boldsymbol{x}, y)$.

However, in the Pconf classification scenario, we are only given positive data equipped with *confidence* $\mathcal{X} := \{(\boldsymbol{x}_i, r_i)\}_{i=1}^n$, where $\boldsymbol{x}_i$ is a positive pattern drawn independently from $p(\boldsymbol{x}|y = +1)$ and $r_i$ is the positive confidence given by $r_i = p(y = +1|\boldsymbol{x}_i)$. Note that this equality does not have to strictly hold as later shown in Section 4. Since we have no access to negative data in the Pconf classification scenario, we cannot directly employ the standard ERM approach. In the next section, we show how the classification risk can be estimated only from Pconf data.

## 3 Pconf classification

In this section, we propose an ERM framework for Pconf classification and derive an estimation error bound for the proposed method. Finally we give examples of practical implementations.

### 3.1 Empirical risk minimization (ERM) framework

Let $\pi_+ = p(y = +1)$ and $r(\boldsymbol{x}) = p(y = +1|\boldsymbol{x})$, and let $\mathbb{E}_+$ denote the expectation over $p(\boldsymbol{x}|y = +1)$. Then the following theorem holds, which forms the basis of our approach:

**Theorem 1.** *The classification risk* (1) *can be expressed as*

$$R(g) = \pi_+ \mathbb{E}_+ \left[ \ell\big(g(\boldsymbol{x})\big) + \frac{1 - r(\boldsymbol{x})}{r(\boldsymbol{x})} \ell\big(-g(\boldsymbol{x})\big) \right], \tag{2}$$

*if we have* $p(y = +1|\boldsymbol{x}) \neq 0$ *for all* $\boldsymbol{x}$ *sampled from* $p(\boldsymbol{x})$.

A proof is given in Appendix A.1 in the supplementary material. Equation (2) does not include the expectation over negative data, but only includes the expectation over positive data and their confidence values. Furthermore, when (2) is minimized with respect to $g$, unknown $\pi_+$ is a proportional constant and thus can be safely ignored. Conceptually, the assumption of $p(y = +1|\boldsymbol{x}) \neq 0$ is implying that the support of the negative distribution is the same or is included in the support of the positive distribution.

Based on this, we propose the following ERM framework for Pconf classification:

$$\min_g \sum_{i=1}^n \left[ \ell\big(g(\boldsymbol{x}_i)\big) + \frac{1 - r_i}{r_i} \ell\big(-g(\boldsymbol{x}_i)\big) \right]. \tag{3}$$

It might be tempting to consider a similar empirical formulation as follows:

$$\min_g \sum_{i=1}^n \left[ r_i \ell\big(g(\boldsymbol{x}_i)\big) + (1 - r_i)\ell\big(-g(\boldsymbol{x}_i)\big) \right]. \tag{4}$$

Equation (4) means that we weigh the positive loss with positive-confidence $r_i$ and the negative loss with negative-confidence $1 - r_i$. This is quite natural and may look straightforward at a glance. However, if we simply consider the population version of the objective function of (4), we have

$$\mathbb{E}_+ \left[ r(\boldsymbol{x})\ell\big(g(\boldsymbol{x})\big) + \big(1 - r(\boldsymbol{x})\big)\ell\big(-g(\boldsymbol{x})\big) \right]$$

$$= \mathbb{E}_+ \left[ p(y = +1|\boldsymbol{x})\ell\big(g(\boldsymbol{x})\big) + p(y = -1|\boldsymbol{x})\ell\big(-g(\boldsymbol{x})\big) \right]$$

$$= \mathbb{E}_+ \left[ \sum_{y \in \{\pm 1\}} p(y|\boldsymbol{x})\ell\big(yg(\boldsymbol{x})\big) \right] = \mathbb{E}_+ \left[ \mathbb{E}_{p(y|\boldsymbol{x})} \big[ \ell\big(yg(\boldsymbol{x})\big) \big] \right], \tag{5}$$

which is *not* equivalent to the classification risk $R(g)$ defined by (1). If the outer expectation was over $p(\boldsymbol{x})$ instead of $p(\boldsymbol{x}|y = +1)$ in (5), then it would be equal to (1). This implies that if we had a different problem setting of having positive confidence equipped for $\boldsymbol{x}$ sampled from $p(\boldsymbol{x})$, this would be trivially solved by a naive weighting idea.

From this viewpoint, (3) can be regarded as an application of *importance sampling* [13, 48] to (4) to cope with the distribution difference between $p(\boldsymbol{x})$ and $p(\boldsymbol{x}|y=+1)$, but with the advantage of *not* requiring training data from the test distribution $p(\boldsymbol{x})$.

In summary, our ERM formulation of (3) is different from naive confidence-weighted classification of (4). We further show in Section 3.2 that the minimizer of (3) converges to the true risk minimizer, while the minimizer of (4) converges to a different quantity and hence learning based on (4) is inconsistent.

## 3.2 Theoretical analysis

Here we derive an estimation error bound for the proposed method. To begin with, let $\mathcal{G}$ be our function class for ERM. Assume there exists $C_g > 0$ such that $\sup_{g \in \mathcal{G}} \|g\|_\infty \leq C_g$ as well as $C_\ell > 0$ such that $\sup_{|z| \leq C_g} \ell(z) \leq C_\ell$. The existence of $C_\ell$ may be guaranteed for all reasonable $\ell$ given a reasonable $\mathcal{G}$ in the sense that $C_g$ exists. As usual [31], assume $\ell(z)$ is Lipschitz continuous for all $|z| \leq C_g$ with a (not necessarily optimal) Lipschitz constant $L_\ell$.

Denote by $\widehat{R}(g)$ the objective function of (3) times $\pi_+$, which is unbiased in estimating $R(g)$ in (1) according to Theorem 1. Subsequently, let $g^* = \arg\min_{g \in \mathcal{G}} R(g)$ be the true risk minimizer, and $\hat{g} = \arg\min_{g \in \mathcal{G}} \widehat{R}(g)$ be the empirical risk minimizer, respectively. The estimation error is defined as $R(\hat{g}) - R(g^*)$, and we are going to bound it from above.

In Theorem 1, $(1 - r(\boldsymbol{x}))/r(\boldsymbol{x})$ is playing a role inside the expectation, for the fact that

$$r(\boldsymbol{x}) = p(y = +1 \mid \boldsymbol{x}) > 0 \text{ for } \boldsymbol{x} \sim p(\boldsymbol{x} \mid y = +1).$$

In order to derive any error bound based on statistical learning theory, we should ensure that $r(\boldsymbol{x})$ could never be too close to zero. To this end, assume there is $C_r > 0$ such that $r(\boldsymbol{x}) \geq C_r$ almost surely. We may trim $r(\boldsymbol{x})$ and then analyze the bounded but biased version of $\widehat{R}(g)$ alternatively. For simplicity, only the unbiased version is involved after assuming $C_r$ exists.

**Lemma 2.** *For any $\delta > 0$, the following uniform deviation bound holds with probability at least $1 - \delta$ (over repeated sampling of data for evaluating $\widehat{R}(g)$):*

$$\sup_{g \in \mathcal{G}} |\widehat{R}(g) - R(g)| \leq 2\pi_+ \left( L_\ell + \frac{L_\ell}{C_r} \right) \mathfrak{R}_n(\mathcal{G}) + \pi_+ \left( C_\ell + \frac{C_\ell}{C_r} \right) \sqrt{\frac{\ln(2/\delta)}{2n}}, \qquad (6)$$

*where $\mathfrak{R}_n(\mathcal{G})$ is the Rademacher complexity of $\mathcal{G}$ for $\mathcal{X}$ of size $n$ drawn from $p(\boldsymbol{x} \mid y = +1)$.[1]*

Lemma 2 guarantees that with high probability $\widehat{R}(g)$ concentrates around $R(g)$ for all $g \in \mathcal{G}$, and the degree of such concentration is controlled by $\mathfrak{R}_n(\mathcal{G})$. Based on this lemma, we are able to establish an estimation error bound, as follows:

**Theorem 3.** *For any $\delta > 0$, with probability at least $1 - \delta$ (over repeated sampling of data for training $\hat{g}$), we have*

$$R(\hat{g}) - R(g^*) \leq 4\pi_+ \left( L_\ell + \frac{L_\ell}{C_r} \right) \mathfrak{R}_n(\mathcal{G}) + 2\pi_+ \left( C_\ell + \frac{C_\ell}{C_r} \right) \sqrt{\frac{\ln(2/\delta)}{2n}}. \qquad (7)$$

Theorem 3 guarantees learning with (3) is consistent [25]: $n \to \infty$ always means $R(\hat{g}) \to R(g^*)$. Consider linear-in-parameter models defined by

$$\mathcal{G} = \{g(\boldsymbol{x}) = \langle w, \phi(\boldsymbol{x}) \rangle_{\mathcal{H}} \mid \|w\|_{\mathcal{H}} \leq C_w, \|\phi(\boldsymbol{x})\|_{\mathcal{H}} \leq C_\phi\},$$

where $\mathcal{H}$ is a Hilbert space, $\langle \cdot, \cdot \rangle_{\mathcal{H}}$ is the inner product in $\mathcal{H}$, $w \in \mathcal{H}$ is the normal, $\phi : \mathbb{R}^d \to \mathcal{H}$ is a feature map, and $C_w > 0$ and $C_\phi > 0$ are constants [43]. It is known that $\mathfrak{R}_n(\mathcal{G}) \leq C_w C_\phi / \sqrt{n}$ [31] and thus $R(\hat{g}) \to R(g^*)$ in $\mathcal{O}_p(1/\sqrt{n})$, where $\mathcal{O}_p$ denotes the order in probability. This order is already the optimal parametric rate and cannot be improved without additional strong assumptions

on $p(\boldsymbol{x}, y)$, $\ell$ and $\mathcal{G}$ jointly [28]. Additionally, if $\ell$ is strictly convex we have $\hat{g} \to g^*$, and if the aforementioned $\mathcal{G}$ is used $\hat{g} \to g^*$ in $\mathcal{O}_p(1/\sqrt{n})$ [3].

At first glance, learning with (4) is numerically more stable; however, it is generally inconsistent, especially when $g$ is linear in parameters and $\ell$ is strictly convex. Denote by $\widehat{R}'(g)$ the objective function of (4) times $\pi_+$, which is unbiased to $R'(g) = \pi_+ \mathbb{E}_+ \mathbb{E}_{p(y|\boldsymbol{x})}[\ell(yg(\boldsymbol{x}))]$ rather than $R(g)$. By the same technique for proving (6) and (7), it is not difficult to show that with probability at least $1 - \delta$,

$$\sup_{g \in \mathcal{G}} |\widehat{R}'(g) - R'(g)| \le 4\pi_+ L_\ell \mathfrak{R}_n(\mathcal{G}) + 2\pi_+ C_\ell \sqrt{\frac{\ln(2/\delta)}{2n}},$$

and hence

$$R'(\hat{g}') - R'(g'^*) \le 8\pi_+ L_\ell \mathfrak{R}_n(\mathcal{G}) + 4\pi_+ C_\ell \sqrt{\frac{\ln(2/\delta)}{2n}},$$

where

$$g'^* = \arg\min_{g \in \mathcal{G}} R'(g) \quad \text{and} \quad \hat{g}' = \arg\min_{g \in \mathcal{G}} \widehat{R}'(g).$$

As a result, when the strict convexity of $R'(g)$ and $\widehat{R}'(g)$ is also met, we have $\hat{g}' \to g'^*$. This demonstrates the inconsistency of learning with (4), since $R'(g) \ne R(g)$ which leads to $g'^* \ne g^*$ given any reasonable $\mathcal{G}$.

### 3.3 Implementation

Finally we give examples of implementations. As a classifier $g$, let us consider a linear-in-parameter model $g(\boldsymbol{x}) = \boldsymbol{\alpha}^\top \boldsymbol{\phi}(\boldsymbol{x})$, where $^\top$ denotes the transpose, $\boldsymbol{\phi}(\boldsymbol{x})$ is a vector of basis functions, and $\boldsymbol{\alpha}$ is a parameter vector. Then from (3), the $\ell_2$-regularized ERM is formulated as

$$\min_{\boldsymbol{\alpha}} \sum_{i=1}^{n} \left[ \ell(\boldsymbol{\alpha}^\top \boldsymbol{\phi}(\boldsymbol{x}_i)) + \frac{1 - r_i}{r_i} \ell(-\boldsymbol{\alpha}^\top \boldsymbol{\phi}(\boldsymbol{x}_i)) \right] + \frac{\lambda}{2} \boldsymbol{\alpha}^\top \boldsymbol{R} \boldsymbol{\alpha},$$

where $\lambda$ is a non-negative constant and $\boldsymbol{R}$ is a positive semi-definite matrix. In practice, we can use any loss functions such as squared loss $\ell_S(z) = (z - 1)^2$, hinge loss $\ell_H(z) = \max(0, 1 - z)$, and ramp loss $\ell_R(z) = \min(1, \max(0, 1 - z))$. In the experiments in Section 4, we use the logistic loss $\ell_L(z) = \log(1 + e^{-z})$, which yields,

$$\min_{\boldsymbol{\alpha}} \sum_{i=1}^{n} \left[ \log(1 + e^{-\boldsymbol{\alpha}^\top \boldsymbol{\phi}(\boldsymbol{x}_i)}) + \frac{1 - r_i}{r_i} \log(1 + e^{\boldsymbol{\alpha}^\top \boldsymbol{\phi}(\boldsymbol{x}_i)}) \right] + \frac{\lambda}{2} \boldsymbol{\alpha}^\top \boldsymbol{R} \boldsymbol{\alpha}. \tag{8}$$

The above objective function is continuous and differentiable, and therefore optimization can be efficiently performed, for example, by quasi-Newton [35] or stochastic gradient methods [45].

## 4 Experiments

In this section, we numerically illustrate the behavior of the proposed method on synthetic datasets for linear models. We further demonstrate the usefulness of the proposed method on benchmark datasets for deep neural networks that are highly nonlinear models. The implementation is based on PyTorch [37], Sklearn [38], and mpmath [20]. Our code will be available on http://github.com/takashiishida/pconf.

### 4.1 Synthetic experiments with linear models

**Setup:** We used two-dimensional Gaussian distributions with means $\boldsymbol{\mu}_+$ and $\boldsymbol{\mu}_-$ and covariance matrices $\boldsymbol{\Sigma}_+$ and $\boldsymbol{\Sigma}_-$, for $p(\boldsymbol{x}|y = +1)$ and $p(\boldsymbol{x}|y = -1)$, respectively. For these parameters, we tried various combinations visually shown in Figure 2. The specific parameters used for each setup are:

- Setup A: $\boldsymbol{\mu}_+ = [0, 0]^\top$, $\boldsymbol{\mu}_- = [-2, 5]^\top$, $\boldsymbol{\Sigma}_+ = \begin{bmatrix} 7 & -6 \\ -6 & 7 \end{bmatrix}$, $\boldsymbol{\Sigma}_- = \begin{bmatrix} 2 & 0 \\ 0 & 2 \end{bmatrix}$.

**Figure 2:** Illustrations based on a single trail of the four setups used in experiments with various Gaussian distributions. The red and green lines are decision boundaries obtained by Pconf and Weighted classification, respectively, where only positive data with confidence are used (no negative data). The black boundary is obtained by O-SVM, which uses only hard-labeled positive data. The blue boundary is obtained by the fully-supervised method using data from both classes. Histograms of confidence of positive data are shown below.

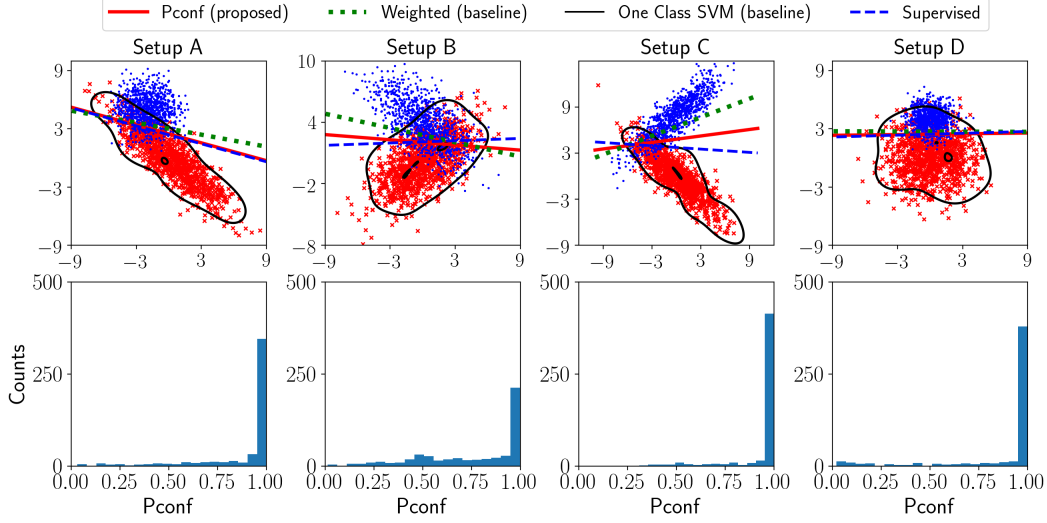

- Setup B:    $\boldsymbol{\mu}_+ = [0,0]^\top, \boldsymbol{\mu}_- = [0,4]^\top, \boldsymbol{\Sigma}_+ = \begin{bmatrix} 5 & 3 \\ 3 & 5 \end{bmatrix}, \boldsymbol{\Sigma}_- = \begin{bmatrix} 5 & -3 \\ -3 & 5 \end{bmatrix}.$

- Setup C:    $\boldsymbol{\mu}_+ = [0,0]^\top, \boldsymbol{\mu}_- = [0,8]^\top, \boldsymbol{\Sigma}_+ = \begin{bmatrix} 7 & -6 \\ -6 & 7 \end{bmatrix}, \boldsymbol{\Sigma}_- = \begin{bmatrix} 7 & 6 \\ 6 & 7 \end{bmatrix}.$

- Setup D:    $\boldsymbol{\mu}_+ = [0,0]^\top, \boldsymbol{\mu}_- = [0,4]^\top, \boldsymbol{\Sigma}_+ = \begin{bmatrix} 4 & 0 \\ 0 & 4 \end{bmatrix}, \boldsymbol{\Sigma}_- = \begin{bmatrix} 1 & 0 \\ 0 & 1 \end{bmatrix}.$

In the case of using two Gaussian distributions, $p(y = +1|\boldsymbol{x}) > 0$ is satisfied for any $\boldsymbol{x}$ sampled from $p(\boldsymbol{x})$, which is a necessary condition for applying Theorem 1. 500 positive data and 500 negative data were generated independently from each distribution for training.[2] Similarly, 1,000 positive and 1,000 negative data were generated for testing. We compared our proposed method (3) with the weighted classification method (4), a regression based method (predict the confidence value itself and post-process output to a binary signal by comparing it to 0.5), one-class support vector machine (O-SVM, [42]) with the Gaussian kernel, and a fully-supervised method based on the empirical version of (1). Note that the proposed method, weighted method, and regression based method only use Pconf data, O-SVM only uses (hard-labeled) positive data, and the fully-supervised method uses both positive and negative data.

In the proposed, weighted, fully-supervised methods, linear-in-input model $g(\boldsymbol{x}) = \boldsymbol{\alpha}^\top \boldsymbol{x} + b$ and the logistic loss were commonly used and *vanilla gradient descent* with $5,000$ epochs (full-batch size) and learning rate $0.001$ was used for optimization. For the regression-based method, we used the squared loss and analytical solution [15]. For the purpose of clear comparison of the risk, we did not use regularization in this toy experiment. An exception was O-SVM, where the user is required to subjectively pre-specify regularization parameter $\nu$ and Gaussian bandwidth $\gamma$. We set them at $\nu = 0.05$ and $\gamma = 0.1$.[3]

**Analysis with true positive-confidence:**    Our first experiments were conducted when true positive-confidence was known. The positive-confidence $r(\boldsymbol{x})$ was analytically computed from the two

**Table 1:** Comparison of the proposed Pconf classification with other methods, with varying degrees of overlap between the positive and negative distributions. We report the mean and standard deviation of the classification accuracy over 20 trials. We show the best and equivalent methods based on the 5% t-test in bold, excluding the fully-supervised method and O-SVM whose settings are different from the others.

| Setup | Pconf | Weighted | Regression | O-SVM | Supervised |
|-------|-------|----------|------------|-------|------------|
| A | **89.7 ± 0.6** | 88.7 ± 1.2 | 68.4 ± 6.5 | 76.0 ± 3.5 | 89.8 ± 0.7 |
| B | **81.2 ± 1.1** | 78.1 ± 1.8 | 73.2 ± 3.2 | 71.3 ± 2.3 | 81.4 ± 1.0 |
| C | **90.2 ± 9.1** | 82.7 ± 13.1 | 50.5 ± 1.7 | 90.8 ± 1.2 | 93.6 ± 0.5 |
| D | **91.5 ± 0.5** | 90.8 ± 0.7 | 64.6 ± 5.3 | 57.1 ± 4.8 | 91.4 ± 0.5 |

**Table 2:** Mean and standard deviation of the classification accuracy with noisy positive confidence. The experimental setup is the same as Table 1, except that positive confidence scores for positive data are noisy. Std. is the standard deviation of Gaussian noise.

| | Setup A | | | | Setup C | |
|---|---------|---|---|---|---------|---|
| Std. | Pconf | Weighted | | Std. | Pconf | Weighted |
| 0.01 | **89.8 ± 0.6** | 88.8 ± 0.9 | | 0.01 | **92.4 ± 1.7** | 84.0 ± 8.2 |
| 0.05 | **89.7 ± 0.6** | 88.3 ± 1.1 | | 0.05 | **92.2 ± 3.3** | 78.5 ± 11.3 |
| 0.10 | **89.2 ± 0.7** | 87.6 ± 1.4 | | 0.10 | **90.8 ± 9.5** | 72.6 ± 12.9 |
| 0.20 | **85.9 ± 2.5** | **85.8 ± 2.5** | | 0.20 | **88.0 ± 9.5** | 65.5 ± 13.1 |

| | Setup B | | | | Setup D | |
|---|---------|---|---|---|---------|---|
| Std. | Pconf | Weighted | | Std. | Pconf | Weighted |
| 0.01 | **81.2 ± 0.9** | 78.2 ± 1.4 | | 0.01 | **91.6 ± 0.5** | 90.6 ± 0.9 |
| 0.05 | **80.7 ± 2.3** | 78.1 ± 1.4 | | 0.05 | **91.5 ± 0.5** | 89.9 ± 1.2 |
| 0.10 | **80.8 ± 1.2** | 77.8 ± 1.5 | | 0.10 | **90.8 ± 0.7** | 88.7 ± 1.8 |
| 0.20 | **77.8 ± 1.4** | **77.2 ± 1.9** | | 0.20 | **87.7 ± 0.8** | 85.5 ± 3.7 |

Gaussian densities and given to each positive data. The results in Table 1 show that the proposed Pconf method is significantly better than the baselines in all cases. In most cases, the proposed Pconf method has similar accuracy compared with the fully supervised case, excluding Setup C where there is a few percent loss. Note that the naive weighted method is consistent if the model is correctly specified, but becomes inconsistent if misspecified [48].[4]

**Analysis with noisy positive-confidence:** In the above toy experiments, we assumed that true positive confidence $r(\boldsymbol{x}) = p(y = +1|\boldsymbol{x})$ is exactly accessible, but this can be unrealistic in practice. To investigate the influence of noise in positive-confidence, we conducted experiments with noisy positive-confidence.

As noisy positive confidence, we added zero-mean Gaussian noise with standard deviation chosen from {0.01, 0.05, 0.1, 0.2}. As the standard deviation gets larger, more noise will be incorporated into positive-confidence. When the modified positive-confidence was over 1 or below 0.01, we clipped it to 1 or rounded up to 0.01 respectively.

The results are shown in Table 2. As expected, the performance starts to deteriorate as the confidence becomes more noisy (i.e., as the standard deviation of Gaussian noise is larger), but the proposed method still works reasonably well in almost all cases.

## 4.2 Benchmark experiments with neural network models

Here, we use more realistic benchmark datasets and more flexible neural network models for experiments.

**Fashion-MNIST:** The *Fashion-MNIST dataset*[5] consists of 70,000 examples where each sample is a $28 \times 28$ gray-scale image (input dimension is 784), associated with a label from 10 fashion item classes. We standardized the data to have zero mean and unit variance.

First, we chose "T-shirt/top" as the positive class, and another item for the negative class. The binary dataset was then divided into four sub-datasets: a training set, a validation set, a test set, and a dataset for learning a probabilistic classifier to estimate positive-confidence. Note that we ask labelers for positive-confidence values in real-world Pconf classification, but we obtained positive-confidence values through a probabilistic classifier here.

We used logistic regression with the same network architecture as a probabilistic classifier to generate confidence.[6] However, instead of weight decay, we used *dropout* [47] with rate 50% after each fully-connected layer, and early-stopping with 20 epochs, since softmax output of flexible neural networks tends to be extremely close to 0 or 1 [14], which is not suitable as a representation of confidence. Furthermore, we rounded up positive confidence less than 1% to 1% to stabilize the optimization process.

We compared Pconf classification (3) with weighted classification (4) and fully-supervised classification based on the empirical version of (1). We used the logistic loss for these methods. We also compared our method with Auto-Encoder [17] as a one-class classification method.

Except Auto-Encoder, we used a fully-connected neural network of three hidden layers ($d$-100-100-100-1) with *rectified linear units (ReLU)* [32] as the activation functions, and weight decay candidates were chosen from $\{10^{-7}, 10^{-4}, 10^{-1}\}$. *Adam* [22] was again used for optimization with 200 epochs and mini-batch size 100.

To select hyper-parameters with validation data, we used the zero-one loss versions of (3) and (4) for Pconf classification and weighted classification, respectively, since no negative data were available in the validation process and thus we could not directly use the classification accuracy. On the other hand, the classification accuracy was directly used for hyper-parameter tuning of the fully-supervised method, which is extremely advantageous. We reported the test accuracy of the model with the best validation score out of all epochs.

Auto-Encoder was trained with (hard-labeled) positive data, and we classified test data into positive class if the mean squared error (MSE) is below a threshold of 70% quantile, and into negative class otherwise. Since we have no negative data for validating hyper-parameters, we sort the MSEs of training positive data in ascending order. We set the weight decay to $10^{-4}$. The architecture is $d$-100-100-100-100 for encoding and the reversed version for decoding, with *ReLU* after hidden layers and *Tanh* after the final layer.

**CIFAR-10:** The *CIFAR-10 dataset* [7] consists of 10 classes, with 5,000 images in each class. Each image is given in a $32 \times 32 \times 3$ format. We chose "airplane" as the positive class and one of the other classes as the negative class in order to construct a dataset for binary classification. We used the neural network architecture specified in Appendix B.1.

For the probabilistic classifier, the same architecture as that for Fashion-MNIST was used except *dropout* with rate 50% was added after the first two fully-connected layers. For Auto-Encoder, the MSE threshold was set to 80% quantile, and we used the architecture specified in Appendix B.2. Other details such as the loss function and weight-decay follow the same setup as the Fashion-MNIST experiments.

**Results:** The results in Table 3 and Table 4 show that in most cases, Pconf classification either outperforms or is comparable to the weighted classification baseline, outperforms Auto-Encoder, and is even comparable to the fully-supervised method in some cases.

**Table 3:** Mean and standard deviation of the classification accuracy over 20 trials for the Fashion-MNIST dataset with fully-connected three hidden-layer neural networks. Pconf classification was compared with the baseline Weighted classification method, Auto-Encoder method and fully-supervised method, with *T-shirt* as the positive class and different choices for the negative class. The best and equivalent methods are shown in bold based on the 5% t-test, excluding the Auto-Encoder method and fully-supervised method.

| P / N | Pconf | Weighted | Auto-Encoder | Supervised |
|---|---|---|---|---|
| T-shirt / trouser | $\mathbf{92.14 \pm 4.06}$ | $85.30 \pm 9.07$ | $71.06 \pm 1.00$ | $98.98 \pm 0.16$ |
| T-shirt / pullover | $\mathbf{96.00 \pm 0.29}$ | $\mathbf{96.08 \pm 1.05}$ | $70.27 \pm 1.22$ | $96.17 \pm 0.34$ |
| T-shirt / dress | $\mathbf{91.52 \pm 1.14}$ | $89.31 \pm 1.08$ | $53.82 \pm 0.93$ | $96.56 \pm 0.34$ |
| T-shirt / coat | $\mathbf{98.12 \pm 0.33}$ | $\mathbf{98.13 \pm 1.12}$ | $68.74 \pm 0.98$ | $98.44 \pm 0.13$ |
| T-shirt / sandal | $\mathbf{99.55 \pm 0.22}$ | $87.83 \pm 18.79$ | $82.02 \pm 0.49$ | $99.93 \pm 0.09$ |
| T-shirt / shirt | $\mathbf{83.70 \pm 0.46}$ | $\mathbf{83.60 \pm 0.65}$ | $57.76 \pm 0.55$ | $85.57 \pm 0.69$ |
| T-shirt / sneaker | $\mathbf{89.86 \pm 13.32}$ | $58.26 \pm 14.27$ | $83.70 \pm 0.26$ | $100.00 \pm 0.00$ |
| T-shirt / bag | $\mathbf{97.56 \pm 0.99}$ | $95.34 \pm 1.00$ | $82.79 \pm 0.70$ | $99.02 \pm 0.29$ |
| T-shirt / ankle boot | $\mathbf{98.84 \pm 1.43}$ | $88.87 \pm 7.86$ | $85.07 \pm 0.37$ | $99.76 \pm 0.07$ |

**Table 4:** Mean and standard deviation of the classification accuracy over 20 trials for the CIFAR-10 dataset with convolutional neural networks. Pconf classification was compared with the baseline Weighted classification method, Auto-Encoder method and fully-supervised method, with *airplane* as the positive class and different choices for the negative class. The best and equivalent methods are shown in bold based on the 5% t-test, excluding the Auto-Encorder method and fully-supervised method.

| P / N | Pconf | Weighted | Auto-Encoder | Supervised |
|---|---|---|---|---|
| airplane / automobile | $\mathbf{82.68 \pm 1.89}$ | $76.21 \pm 2.43$ | $75.13 \pm 0.42$ | $93.96 \pm 0.58$ |
| airplane / bird | $\mathbf{82.23 \pm 1.21}$ | $80.66 \pm 1.60$ | $54.83 \pm 0.39$ | $87.76 \pm 4.97$ |
| airplane / cat | $85.18 \pm 1.35$ | $\mathbf{89.60 \pm 0.92}$ | $61.03 \pm 0.59$ | $92.90 \pm 0.58$ |
| airplane / deer | $\mathbf{87.68 \pm 1.36}$ | $\mathbf{87.24 \pm 1.58}$ | $55.60 \pm 0.53$ | $93.35 \pm 0.77$ |
| airplane / dog | $\mathbf{89.91 \pm 0.85}$ | $\mathbf{89.08 \pm 1.95}$ | $62.64 \pm 0.63$ | $94.61 \pm 0.45$ |
| airplane / frog | $\mathbf{90.80 \pm 0.98}$ | $81.84 \pm 3.92$ | $62.52 \pm 0.68$ | $95.95 \pm 0.40$ |
| airplane / horse | $\mathbf{89.82 \pm 1.07}$ | $85.10 \pm 2.61$ | $67.55 \pm 0.73$ | $95.65 \pm 0.37$ |
| airplane / ship | $\mathbf{69.71 \pm 2.37}$ | $\mathbf{70.68 \pm 1.45}$ | $52.09 \pm 0.42$ | $81.45 \pm 8.87$ |
| airplane / truck | $81.76 \pm 2.09$ | $\mathbf{86.74 \pm 0.85}$ | $73.74 \pm 0.38$ | $92.10 \pm 0.82$ |

## 5 Conclusion

We proposed a novel problem setting and algorithm for binary classification from positive data equipped with confidence. Our key contribution was to show that an unbiased estimator of the classification risk can be obtained for positive-confidence data, without negative data or even unlabeled data. This was achieved by reformulating the classification risk based on both positive and negative data, to an equivalent expression that only requires positive-confidence data. Theoretically, we established an estimation error bound, and experimentally demonstrated the usefulness of our algorithm.

**Acknowledgments**

TI was supported by Sumitomo Mitsui Asset Management. MS was supported by JST CREST JPMJCR1403. We thank Ikko Yamane and Tomoya Sakai for the helpful discussions. We also thank anonymous reviewers for pointing out numerical issues in our experiments, and for pointing out the necessary condition in Theorem 1 in our earlier work of this paper.

## Footnotes

[1] $\mathfrak{R}_n(\mathcal{G}) = \mathbb{E}_{\mathcal{X}} \mathbb{E}_{\sigma_1,\ldots,\sigma_n} [\sup_{g \in \mathcal{G}} \frac{1}{n} \sum_{\boldsymbol{x}_i \in \mathcal{X}} \sigma_i g(\boldsymbol{x}_i)]$ where $\sigma_1, \ldots, \sigma_n$ are $n$ Rademacher variables following [31].

[2]Negative training data are used only in the fully-supervised method that is tested for performance comparison.

[3]If we naively use default parameters in Sklearn [38] instead, which is the usual case in the real world without negative data for validation, the classification accuracy of O-SVM is worse for all setups except D in Table 1, which demonstrates the difficulty of using O-SVM.

[4]Since our proposed method has coefficient $\frac{1-r_i}{r_i}$ in the 2nd term of (3), it may suffer numerical problems, e.g., when $r_i$ is extremely small. To investigate this, we used the mpmath package [20] to compute the gradient with arbitrary precision. The experimental results were actually not that much different from the ones obtained with single precision, implying that the numerical problems are not much troublesome.

[5]https://github.com/zalandoresearch/fashion-mnist

[6]Both positive and negative data are used to train the probabilistic classifier to estimate confidence, and this data is separated from any other process of experiments.

[7]https://www.cs.toronto.edu/~kriz/cifar.html

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
