[Supplementary Material]

# A  Proofs

## A.1  Proof of Theorem 1

The classification risk (1) can be expressed and decomposed as

$$
R(g) = \sum_{y=\pm 1} \int \ell(yg(\boldsymbol{x}))p(\boldsymbol{x}|y)p(y)\mathrm{d}\boldsymbol{x}
$$

$$
= \int \ell(g(\boldsymbol{x}))p(\boldsymbol{x}|y=+1)p(y=+1)\mathrm{d}\boldsymbol{x} + \int \ell(-g(\boldsymbol{x}))p(\boldsymbol{x}|y=-1)p(y=-1)\mathrm{d}\boldsymbol{x}
$$

$$
= \pi_+ \mathbb{E}_+[\ell(g(\boldsymbol{x}))] + \pi_- \mathbb{E}_-[\ell(-g(\boldsymbol{x}))], \tag{9}
$$

where $\pi_- = p(y=-1)$ and $\mathbb{E}_-$ denotes the expectation over $p(\boldsymbol{x}|y=-1)$. Since

$$
\pi_+ p(\boldsymbol{x}|y=+1) + \pi_- p(\boldsymbol{x}|y=-1) = p(\boldsymbol{x}, y=+1) + p(\boldsymbol{x}, y=-1)
$$

$$
= p(\boldsymbol{x})
$$

$$
= \frac{p(\boldsymbol{x}, y=+1)}{p(y=+1|\boldsymbol{x})}
$$

$$
= \frac{\pi_+ p(\boldsymbol{x}|y=+1)}{r(\boldsymbol{x})},
$$

where the third equality requires the assumption of $p(y=+1|\boldsymbol{x}) \neq 0$ stated in Theorem 1. we have

$$
\pi_- p(\boldsymbol{x}|y=-1) = \pi_+ p(\boldsymbol{x}|y=+1)\left(\frac{1-r(\boldsymbol{x})}{r(\boldsymbol{x})}\right).
$$

Then the second term in (9) can be expressed as

$$
\pi_- \mathbb{E}_-[\ell(-g(\boldsymbol{x}))] = \int \pi_- p(\boldsymbol{x}|y=-1)\ell(-g(\boldsymbol{x}))\mathrm{d}\boldsymbol{x}
$$

$$
= \int \pi_+ p(\boldsymbol{x}|y=+1)\left(\frac{1-r(\boldsymbol{x})}{r(\boldsymbol{x})}\right)\ell(-g(\boldsymbol{x}))\mathrm{d}\boldsymbol{x}
$$

$$
= \pi_+ \mathbb{E}_+\left[\frac{1-r(\boldsymbol{x})}{r(\boldsymbol{x})}\ell(-g(\boldsymbol{x}))\right],
$$

which concludes the proof. $\qquad\square$

## A.2  Proof of Lemma 2

By assumption, it holds almost surely that

$$
\frac{1-r(\boldsymbol{x})}{r(\boldsymbol{x})} \leq \frac{1}{C_r};
$$

due to the existence of $C_\ell$, the change of $\widehat{R}(g)$ will be no more than $(C_\ell + C_\ell/C_r)/n$ if some $\boldsymbol{x}_i$ is replaced with $\boldsymbol{x}_i'$.

Consider a single direction of the uniform deviation: $\sup_{g\in\mathcal{G}} \widehat{R}(g) - R(g)$. Note that the change of $\sup_{g\in\mathcal{G}} \widehat{R}(g) - R(g)$ shares the same upper bound with the change of $\widehat{R}(g)$, and *McDiarmid's inequality* [27] implies that

$$
\Pr\left\{\sup_{g\in\mathcal{G}} \widehat{R}(g) - R(g) - \mathbb{E}_{\mathcal{X}}\left[\sup_{g\in\mathcal{G}} \widehat{R}(g) - R(g)\right] \geq \epsilon\right\} \leq \exp\left(-\frac{2\epsilon^2 n}{(C_\ell + C_\ell/C_r)^2}\right),
$$

or equivalently, with probability at least $1 - \delta/2$,

$$
\sup_{g\in\mathcal{G}} \widehat{R}(g) - R(g) \leq \mathbb{E}_{\mathcal{X}}\left[\sup_{g\in\mathcal{G}} \widehat{R}(g) - R(g)\right] + \left(C_\ell + \frac{C_\ell}{C_r}\right)\sqrt{\frac{\ln(2/\delta)}{2n}}.
$$

Since $\widehat{R}(g)$ is unbiased, it is routine to show that [31]

$$\mathbb{E}_{\mathcal{X}}\left[\sup_{g\in\mathcal{G}}\widehat{R}(g)-R(g)\right] \leq 2\mathfrak{R}_n\left(\left(1+\frac{1-r}{r}\right)\circ\ell\circ\mathcal{G}\right)$$

$$\leq 2\left(1+\frac{1}{C_r}\right)\mathfrak{R}_n(\ell\circ\mathcal{G})$$

$$\leq 2\left(L_\ell+\frac{L_\ell}{C_r}\right)\mathfrak{R}_n(\mathcal{G}),$$

which proves this direction.

The other direction $\sup_{g\in\mathcal{G}}R(g)-\widehat{R}(g)$ can be proven similarly. $\qquad\square$

### A.3 Proof of Theorem 3

Based on Lemma 2, the estimation error bound (7) is proven through

$$R(\hat{g})-R(g^*) = \left(\widehat{R}(\hat{g})-\widehat{R}(g^*)\right)+\left(R(\hat{g})-\widehat{R}(\hat{g})\right)+\left(\widehat{R}(g^*)-R(g^*)\right)$$

$$\leq 0 + 2\sup_{g\in\mathcal{G}}|\widehat{R}(g)-R(g)|$$

$$\leq 4\left(L_\ell+\frac{L_\ell}{C_r}\right)\mathfrak{R}_n(\mathcal{G})+2\left(C_\ell+\frac{C_\ell}{C_r}\right)\sqrt{\frac{\ln(2/\delta)}{2n}},$$

where $\widehat{R}(\hat{g})\leq\widehat{R}(g^*)$ by the definition of $\widehat{R}$. $\qquad\square$

## B Neural Network Architectures used in Section 4.2

### B.1 CNN architecture

- Convolution (3 in- /18 out-channels, kernel size 5).
- Max-pooling (kernel size 2, stride 2).
- Convolution (18 in- /48 out-channels, kernel size 5).
- Max-pooling (kernel size 2, stride 2).
- Fully-connected (800 units) with ReLU.
- Fully-connected (400 units) with ReLU.
- Fully-connected (1 unit).

### B.2 AutoEncoder Architecture

- Convolution (3 in- /18 out-channels, kernel size 5, stride 1) with ReLU.
- Max-pooling (kernel size 2, stride 2).
- Convolutional layer (18 in- /48 out-channels, kernel size 5, stride 1) with ReLU.
- Max-pooling (kernel size 2, stride 2).
- Deconvolution (48 in- /18 out-channels, kernel size 5, stride 2) with ReLU.
- Deconvolution (18 in- /5 out-channels, kernel size 5, stride 2).
- Deconvolution (5 in- /3 out-channels, kernel size 4, stride 1) with Tanh.