[Reviews · NeurIPS 2018]

Reviewer 1



This paper deals with the problem of binary classification in the case where the training sample consists of positive data equiped with confidence (Pconf) The authors show that the classification risk can be expressed in terms of positive data. An empirical risk minimization framework is considered. they derive rates of convergence for the emprical risk minimizer estimator. Futhermore, they perform a simulation study which shows the good performances of the proposed method for classifcation problems. I find the paper interesting and very clear. Hereafter my comments/questions: 1) I am a bit suprised by the fact that the authors do not cite the following paper "Classification from Pairwise Similarity and Unlabeled Data" (Han Bao, Gang Niu, Masashi Sugiyama) Indeed, this paper seems to consider similar techniques for the positive-unlabeled classification framework. 2)Can we improve the rate of convergence with some extra assumptions on the distribution (e.g margin conditions) and on the loss function (e.g on the modulus of convexity) ? Response to the Rebuttal ---------------------------------- I read the rebuttal and the other reviews. I thank the authors for their responses. I update the overall score of the paper.

Reviewer 2



Overview and Recommendation: This paper presents an algorithm and theoretical analysis for Pconf learning of a binary classifier, when only the positive instances and their conditional class probabilities are available. In contrast to PU learning, which has received attention recently in the ML community, Pconf learning does not require a large amount of unlabeled data or any knowledge of class priors, however it does require confidence weights p(y=+1 | x) for each example x sampled from p(x | y= +1). The paper is well written and this solution can be useful for many real world applications, therefore, it is a clear accept. Although, the analysis is very similar to the recent analysis for PU learning published by Du Plessis et al. recently at NIPS but the problem setting is novel and interesting. However, I think that the authors should fix some technical issues with their simulation experiment. Summary: The main contribution of the paper is the observation that empirical risk minimization for Pconf learning can be done by importance weighting the confidence weighted risk (equation 3) using positive samples. The paper also provides an estimatior error bound for the proposed risk-estimator. The analysis only requires a bounded hypothesis class and a bounded lipschitz loss function on that hypothesis set and shows that estimation error decreases as 1/sqrt(n), which is expected and intuitive. The experimental results also support the theory. Technical Issues: Loss of accuracy when the separation between clusters increases should simply not happen. The authors justify this by a loss-of-senstivity in the confidence values but in the end this is just a numerical issue. The gradient of the log-linear model used in the experiment can be coded directly, and then an arbitrary-precision floating-point library for python such as mpmath can be used. Since the dimensionality is just 2 and the number of examples is 1000 this experiment should not take up much resources. ==== After Author Response === I am happy to see that the authors made the effort to run the experiment using the arb prec math library which made the experimental results more intuitive. I have increased my rating of the paper, because now the paper does not seem to have any technical issues.

Reviewer 3



The authors study a new classification setting in this paper, where only positive samples and their confidence to be positive are given. The authors find several interesting applications of the proposed new setting. I think the new setting is meaningful and may interest many readers. Specifically, the authors proposed an unbiased estimator for learning a binary classifier from the positive samples and their confidence. Then, sufficient experiments have been done to empirically show the effectiveness of the proposed method. Overall, the paper is well-organised and well-written. My primary concern is how to obtain the confidence of positive data. Compared with the one class learning setting, obtaining numbers of confidence looks harder than obtaining integer hard labels. I do understand that if we just employ the integer hard labels, some more details are missing, e.g., the ranking information. In the rebuttal, I would like the authors to discuss how to estimate the confidence without the help of negative examples. How to tune the regularisation parameter in the proposed objective function? Since we only have positive examples, the traditional cross-validation method may not apply. We should be every careful to tune the parameter because it is closely related to the class prior information as shown in Eq. (2). For different class priors, the regularisation parameter should be different even if the training sample size is fixed. The authors have set a section to discuss how the overlap between the positive and negative examples affects the performance of the proposed method, which is quiet interesting. However, I think we should focus more on here. What are the covariance matrices of the Gaussians? Can the proposed method still work well when the overlap is huge? ========== I have three major concerns with the paper. In the rebuttal, the authors have well addressed the second and third concerns. Although the authors haven't provided a detailed method to obtain the confidence for positive examples, the proposed unbiased estimator is interesting and can find potential applications. I am satisfied with paper and do not change my score.